# Effect of the Content of Micro-Active Copper Tailing on the Strength and Pore Structure of Cementitious Materials

**DOI:** 10.3390/ma12111861

**Published:** 2019-06-09

**Authors:** Liming Zhang, Songbai Liu, Dongsheng Song

**Affiliations:** 1Jiang Xi Buidings Materials Scientific Research & Design Institute, Nanchang 330001, China; Lsb8811@126.com (S.L.); Sds1791@126.com (D.S.); 2Department of Civil Engineering, Tsinghua University, Beijing 10084, China

**Keywords:** copper tailing, pore structure, microaggregate filling, compressive strength, particle size distribution

## Abstract

This study investigates the effect of micr-oaggregate filling with copper tailing on the pore structure of cement paste containing copper tailing (CPCT). The particle size of the CPCT and the pore structure of CPCT were analyzed by laser particle size analysis and mercury instruction porosimetry (MIP). Results showed that at the early stage of curing time, with increasing copper tailing content, the compressive strength of cement mortar with copper tailing (CMCT) was lower, and the porosity and pore diameter of CPCT were higher and greater; with the extension of curing age, when the content of copper tailing was less than 30%, the compressive strength of CMCT and the porosity of CPCT changed slightly with the increase of the content of copper tailing. However, the maximum hole diameter of CPCT decreased gradually (a curing age between 7 d and 365 d under standard conditions). Scanning electron microscopy analysis showed that at the early stage of cement hydration in the CPCT, the copper tailing did not fill the pores in CPCT well, while in the later stage of cement hydration, the microaggregates of copper tailing filled the pores well and closely combined with the surrounding hydration products. In the later stage of cement hydration, the microaggregate filling of copper tailing was primarily responsible for the strength increase of the CMCT.

## 1. Introduction

The inventory of copper tailings in Jiangxi (Figure 1) is 496.17 million tons [1], with an annual increase of 44 million tons—the highest in Asia. The annual utilization rate is only 1% [2]. A large number of non-exploitable tailing dam piles exist in the tailing dam, which has emerged as a significant potential safety hazard for the local ecological environment, and hence, urgently needs resource utilization [3]. Domestic and international experts [4,5,6,7,8] have focused on the preparation of concrete as a mineral admixture and studied its performance. They found that when the content of copper tailing slag is 5% of the cementing material, the performance of concrete can be improved, but beyond this content, the performance is deteriorated. Such a low content limits the large-scale application of copper tailing as mineral admixture in cement concrete, so it was urgent to find ways to improve the content of copper tailing in cement. Research showed that the pore structure has a significant impact on the mechanical properties and carbonation resistance [9], chloride ion erosion resistance [10,11,12,13], and other durability properties of cement concrete [14,15,16,17,18,19]. However, there is little research on the microscopic mechanism, especially on the pore structure of cement mortar with copper tailing (CMCT). Studying the pore structure is of great significance to understand the relationship between the performance and microstructure of CMCT. 

Hence, in this study, a ball mill was used to grind copper tailing to obtain a specific surface area of 591 m^2^/kg; X-ray Computed Tomography Imaging (X-ct )was used to study the compressive strength of CMCT and pore structure of cement paste containing copper tailing (CPCT) of different ages under standard curing conditions, and scanning electron microscopy (SEM) was used to conduct auxiliary research to explore the influence of microaggregate filling of copper tailing on the pore structure of CPCT, and thereby reveal the micromechanism of cement hydration of copper tailing.

## 2. Materials and Methods 

### 2.1. Raw Materials

Copper tailing were procured from a copper mine in Ruichang City, Jiangxi Province, with the main chemical composition (Table 1) of SiO_2_, Al_2_O_3_, and CaO and trace amounts of Fe_2_O_3_ and MgO. After grinding for 45 min with a special ball mill (Tencan Powder Changsha, Hunnan provice, China), the specific surface area of the copper tailing was 591 m^2^/kg. The particle size distribution was determined by laser particle size analysis, as shown in Figure 2a,b. The chemical composition and particle size distribution of Jiangxi Conch 42.5 Ordinary Portland cement are shown in Table 1 and Figure 2a,b. The sand was standard sand and the particle size distribution of it is shown in Figure 2a,b. The water is tap water.

Figure 2 shows the particle size distribution of the raw materials. Figure 2a shows that the mean particle size of copper tailing admixture after grinding for 45 min was around 5.6 μm with 44% of particles passing, and the particle size range was 0–48.1 μm; the mean particle size of cement was around 17.0 μm with 47%, and the particle size range of cement was 0–100 μm; the mean particle size of standard sand was 500 μm with 58%, and the particle size range of standard sand was 80–2000 μm. The mean particle size of copper tailing was 0.33 and 0.01 times that of cement and standard sand, respectively. The maximum particle size of copper tailing was 0.48 times and 0.02 times that of cement and standard sand, respectively. Therefore, copper tailing can be filled in the pores of ordinary Portland cement [20,21,22,23].

### 2.2. Production of Test Pieces

The cement mortar mix proportion and performance of the specimens is shown in Table 2. The setting time of cement mortar was not affected by the addition of copper tailing, and the fluidity of cement mortar with copper tailing was greater than 195 mm, thereby meeting the requirements of the standard (GB/T1596-2017) [24]. Cement mortar specimens with size 70.7 mm × 70.7 mm × 70.7 mm were subjected to strength testing. Microstructure analysis was conducted with cement pastes of size 5 mm × 5 mm × 5 mm. Twenty four hours after releasing from the mold, the specimens were cured in a standard environment (20 °C, humidity 95% or higher) for 6 d, 27 d, 89 d, 179 d, and 364 d, and were dried to a constant weight after the test. Then the cement mortar was tested for its mechanical intensity, micro-structure tests were conducted on the termination of hydration in anhydrous ethanol. The mixing of copper tailing did not affect the working performance of the cement mortar, as can be seen in Table 2.

### 2.3. Experimental Approach

#### 2.3.1. Compressive Strength Test

The compressive strength of the CMCT specimens was tested in accordance with the method specified in GB/ t1596-2017 “Fly Ash in Cement and Concrete” [24].

#### 2.3.2. X-CT Test

The MicroXCT-400 X-ray 3D reconstruction scanning microscope (Carl Zeiss, Jena, Germany) was used for the X-CT test. For all samples, the X-ray tube voltage and current were 77 kV and 129 mA, respectively, during the text, and the pixel size was 1024 × 1024 with a resolution of 9.65 m. The object rotation angle was 360°, and the exposure time for each projection was 5 s. The hardened paste was taken after reaching the target age, the volume was broken to about 1 cm^3^, hydration was terminated by anhydrous ethanol, and it was vacuum dried at 60 °C until a constant weight was reached.

#### 2.3.3. SEM Observation 

The Hitachi s-4800 (Hitachi, Tokyo, Japan) was used to conduct SEM analysis at an operating voltage of 10.0 kV. As with the X-CT test, the second electron image was used to observe the morphology of hydration products, and the amplification factor was 500 times.

## 3. Results and Discussion

### 3.1. Compressive Strength

Results from the compressive strength test of the CMCT with different curing ages are shown in Figure 3. The compressive strength of each CMCT increased with increasing curing age. With copper tailing content increase, when the curing age was less than 28 d, the compressive strength of CMCT decreased; when the curing age was 28 d, the compressive strength of CMCT with less than 10% copper tailing content was higher than that of ordinary cement mortar; when curing age was between 28 d and 180 d and copper tailing content was between 10% and 30%, the compressive strength of CMCT increased sharply; when the curing age was 180 d, the compressive strength of CMCT with less than 20% copper tailing content was close to that of ordinary cement mortar; when the curing age was between 180 d and 365 d and the copper tailing content was less than 30%, the compressive strength of CMCT exceeded that of ordinary cement mortar. This is because the later hydration product of cement refined the pores of the mortar sample. In the sample, pore size was less than 30 μm, accounting for 80% of the total number of pores, and the particle size of less than 30 μm of copper tailing accounted for 93.1% of the total, which was filled in the cement hydration hole, increased the compactness of cement mortar, and then increased the compressive strength of CMCT.

### 3.2. X-CT Results

Figure 4 shows the three-dimensional reconstruction of X-CT images of the CM30 samples after curing for 28 d, 180 d, and 365 d. The dark areas in the samples are all pores. As is evident, the number of large pores in the samples after curing for 180 d and 365 d decreased significantly, while the number of small holes increased. The pore volume distribution was more concentrated in the small holes, indicating that the extended curing time and cement hydration had a refining effect on the pore diameter in the CPCT.

### 3.3. MIP Results for Hole Structure

#### 3.3.1. Porosity

Figure 5 shows the effect of curing age on the porosity of the cement paste. With extended curing age, the porosity of the CPCT decreased. When the curing age was less than 180 d, the porosity of the CPCT increased with increasing copper tailing content; on the other hand, when the curing age was more than 180 d, the copper tailing content had little influence on the porosity of the CPCT.

#### 3.3.2. Maximum Aperture Distribution

Figure 6 shows the pore size distribution of the hardened slurry with different copper tailing contents. The peak value on the curve is the maximum aperture. The maximum pore diameter of the cement slurry gradually decreased with increasing curing age; this was also observed for the CPCT. With increasing copper tailing content, when the curing age was 7 d, the maximum pore size increased, whereas when the curing age exceeded 28 d, the maximum pore size decreased. This is because of the many early cement hydration products [25], larger pore size [26], less copper tailing in the filling pores, and more capillary pores in CPCT. With prolonged aging, the maximum pore size of the CPCT was smaller than that of the pure cement sample. The results show that the pore size of the cement was decreased by the product of continuous hydration of cement, the capillary pores decreased due to filling by the copper tailing, and the pores of the cement filled with the copper tailing slag microaggregate were more compact. Even with prolonged aging, the above trend, i.e., smaller maximum pore diameter of the CPCT, existed.

### 3.4. SEM

The microstructures of the CPCT at different ages observed by SEM are shown in Figure 7. Figure 7a–c show that, after curing for 7 d at room temperature, the hydration products of pure cement sample C0 were abundant, and were distributed on the surface of unhydrated particles. However, the CPCT structure was not dense enough, and the bond between the hydration products was not strong. For the CM10 and CM30 samples of CPCT, although some hydration products grew or covered the surface of unhydrated particles in the pores, the hydration products were not full and there were many pores. In particular, a large number of smooth rhomboid unhydrated copper tailing particles were observed in the CM30 sample with a high copper tailing content. This indicates that during curing at room temperature for 7d, the copper tailings did not fill the pores in the CPCT well. With the prolongation of the phase, the microstructure of each sample at 28 d and 180 d is shown in Figure 7d–i. With the growth of the phase, the hydration products in each sample were abundant, the accumulation was dense, the CPCT structure was dense, and the pores were few. A large number of flaked Ca(OH)_2_ crystals were observed in samples with 10% and 30% copper tailing content, indicating that the copper tailing did not consume Ca(OH)_2_ and there was a large amount of Ca(OH)_2_ in the slurry. Figure 7j–l show the microstructure of each sample at the long age. All samples of the CPCT were very dense, with abundant hydration products, and good filling pores. Even with the addition of copper tailing, it can be seen that the microaggregates of copper tailing had a good filling effect and closely combined with the surrounding hydration products.

## 4. Conclusions

The compressive strength of CMCT and pore structure of CPCT cured to different ages were studied, and the following conclusions were obtained.The compressive strength of CMCT increased with increasing curing age and was close to that of ordinary cement mortar with different curing time and copper tailing content. The curing time which the CMCT compressive strength was close to ordinary cement mortar compresssive increased with the increase of copper tailing content (less than 30%). When the curing age was 365 d, the copper tailing content was within 30%, and further addition of tailing did not affect the compressive strength of concrete.The porosity of the CPCT decreased with increasing curing age and increased with the increase of copper tailing content when the curing age was less than 180 d; the copper tailing content had little influence on the porosity of the CPCT when the curing age was more than 180 d.The maximum pore diameter of the CPCT gradually decreased with increasing curing age and increased with the increase of copper tailing content when the curing age was less than 7d; with curing time increases, the maximum pore diameter of the CPCT gradually decreased with the increase of copper tailing content. The X-CT test results showed that prolonged curing and cement hydration played a role in refining the pore diameter of the CPCT.In the early stage, the CPCT pore structure was less dense with copper tailing increases. With prolonged curing, the CPCT structure was gradually denser with copper tailing increases; the CPCT pore structure was denser with copper tailing increases until the curing age was more than 365 d. The microaggregates of solid copper tailings had a good filling effect and closely combined with the surrounding hydration products.

## Figures and Tables

**Figure 1 materials-12-01861-f001:**
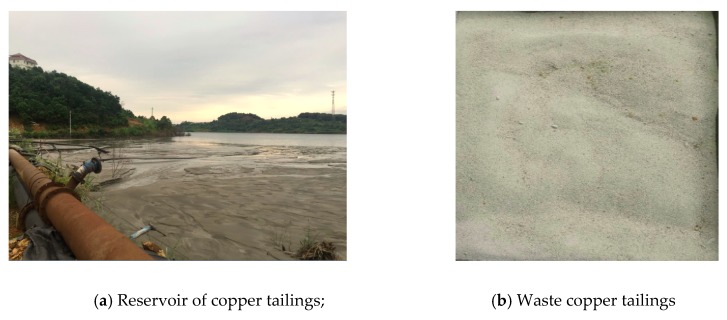
Copper tailings in Jiangxi Province.

**Figure 2 materials-12-01861-f002:**
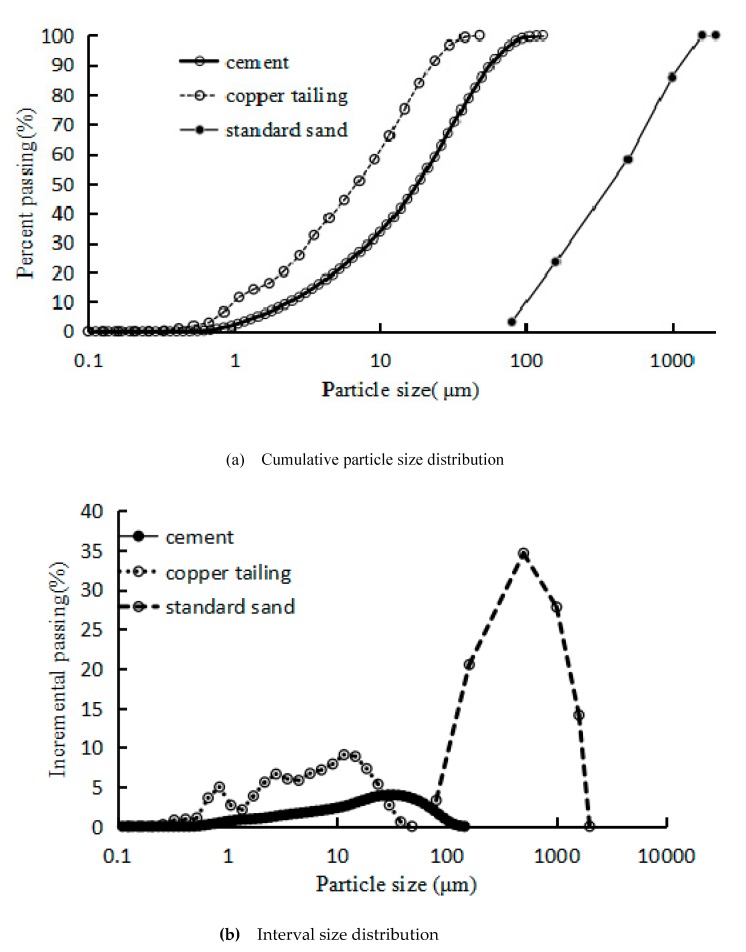
Particle size distribution of raw materials. (**a**) Cumulative particle size distribution; (**b**) Interval size distribution.

**Figure 3 materials-12-01861-f003:**
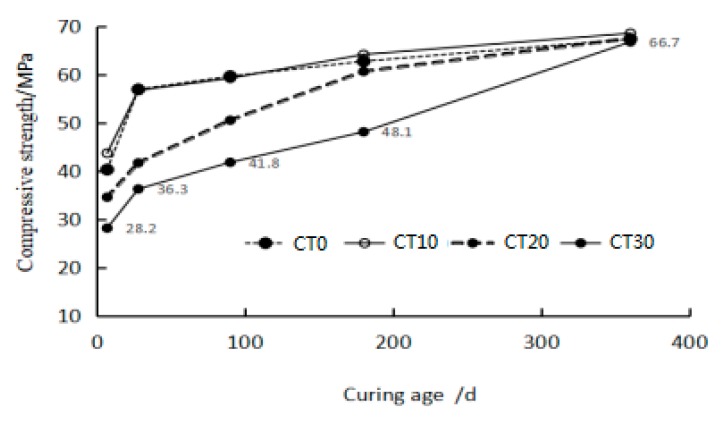
Compressive strength of cement mortar with copper tailing (CMCT) at different curing ages.

**Figure 4 materials-12-01861-f004:**
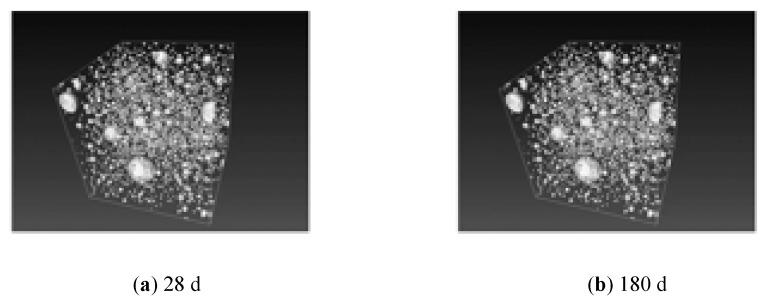
X-CT images of the pores inside the CM30 samples.

**Figure 5 materials-12-01861-f005:**
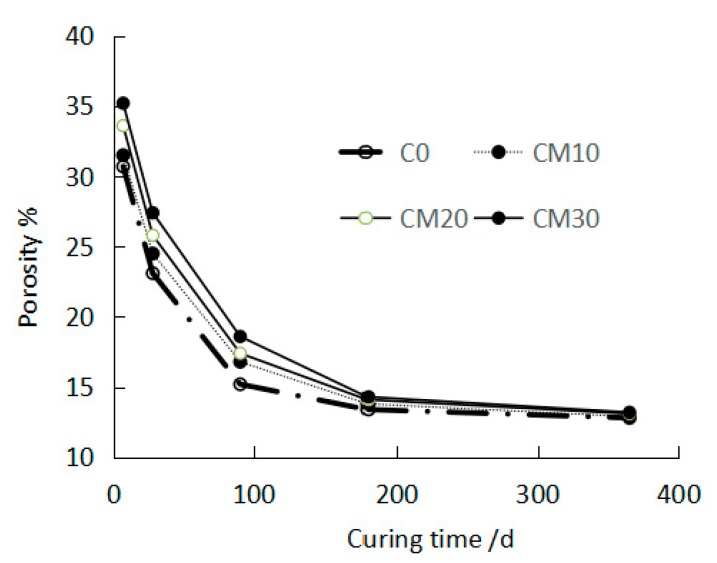
Effect of curing age of cement paste containing copper tailing (CPCT) on porosity.

**Figure 6 materials-12-01861-f006:**
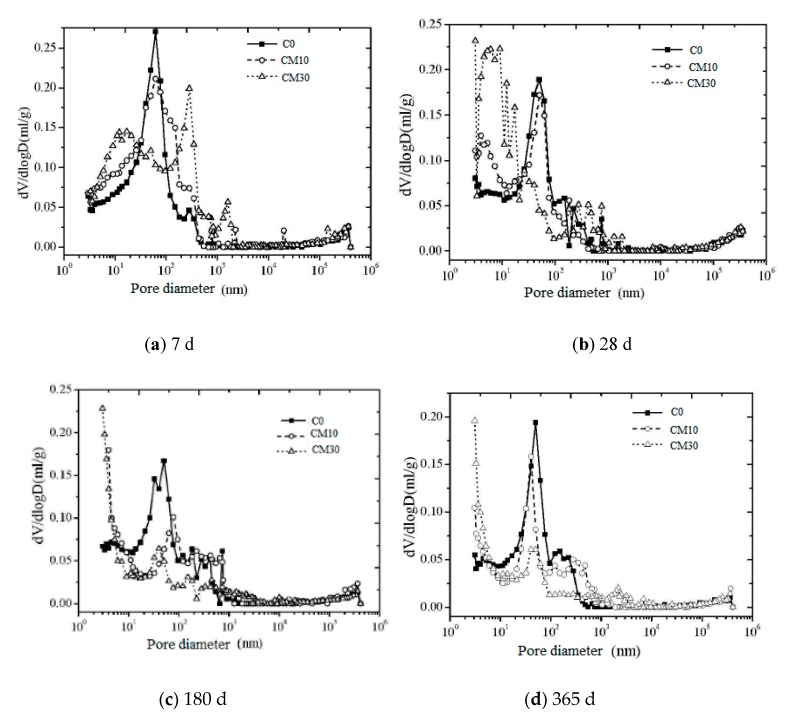
Effect of copper tailing content on maximum pore size of CPCT.

**Figure 7 materials-12-01861-f007:**
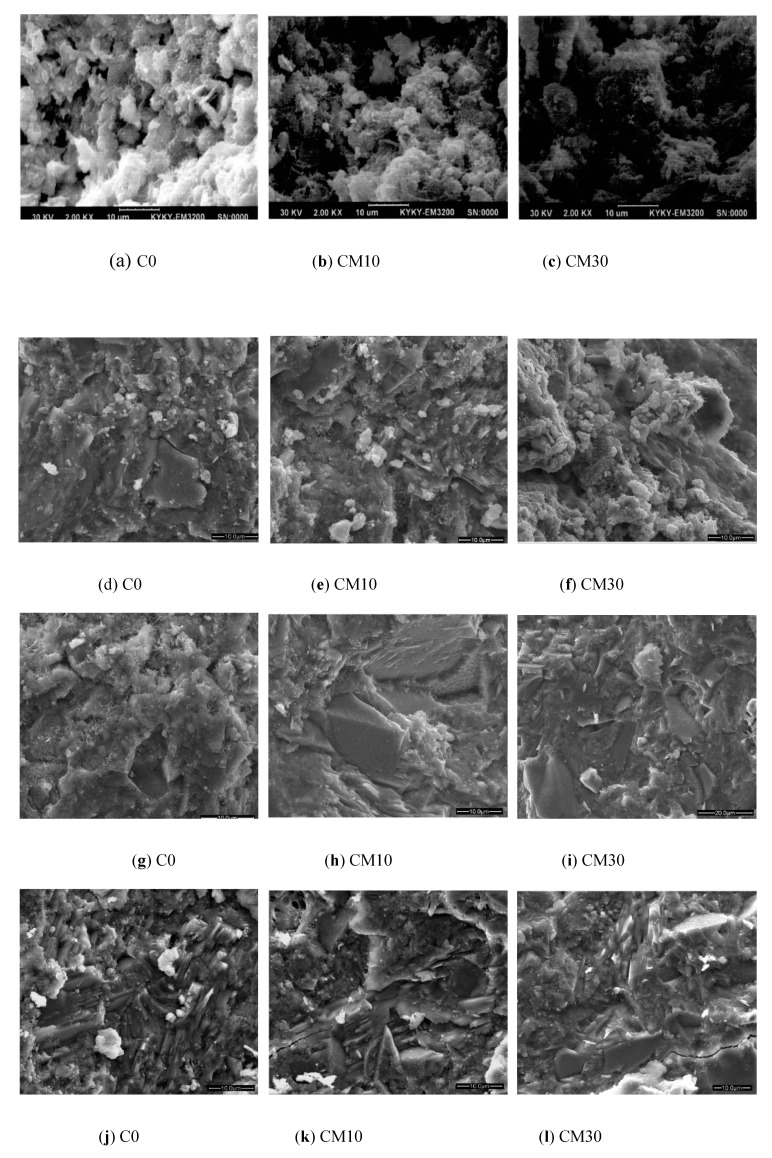
SEM images of composite cementitious material at different curing ages. Curing for 7 d: (**a**) C0; (**b**) CM10; (**c**) CM30; Curing for 28 d: (**d**) C0; (**e**) CM10; (**f**) CM30; Curing for 180 d: (**g**) C0; (**h**) CM10; (**i**) CM30; Curing for 365 d: (**j**) C0; (**k**) CM10; (**l**) CM30.

**Table 1 materials-12-01861-t001:** Chemical composition of cementitious material.

Binder Material Type	SiO_2_	Al_2_O_3_	Fe_2_O_3_	CaO	MgO	SO_3_	Loss
Cement	26.5	6.4	3.3	55.7	1.7	2.0	5.3
Copper tailing slag	38.5	6.6	15.8	32.7	2.8	3.2	0.4

**Table 2 materials-12-01861-t002:** Cement mortar mix proportion and workable performance.

Sample	Cement/g	Copper Tailing/s	Sand/g	Water/g	Initial Setting Time/min	Final Setting Time/min	Fluidity/mm
CT0	450	0	1350	225	170	215	220
CT10	405	45	1350	225	170	215	225
CT20	360	90	1350	225	170	215	213
CT30	315	135	1350	225	170	215	195
C0	100	0	0	25	170	215	-
CM10	90	10	0	25	170	215	-
CM20	80	20	0	25	170	215	-
CM30	70	30	0	25	170	215	-

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
