# Peer review of "Effect of the Content of Micro-Active Copper Tailing on the Strength and Pore Structure of Cementitious Materials"

_materials, 2019, doi:10.3390/ma12111861_

Round 1
Reviewer 1 Report
It is rather a lab test report but a nice piece of work. The paper title should be completed as follows: on the compressive strength and Portland cement mortar.
Table 2 – a misspelling, should be cement not ceme.
Conclusion 1: “further addition of tailings does not affect the compressive strength of concrete”. This statement is doubly unjustified. First of all, the subject of the study is not concrete but mortar. Secondly, the influence of the copper tailings contested above 30% has been not tested.
Conclusion 2: “speed of decrease” better “rate of decrease”. Proof-reading by a native speaker is strongly recommended.
The literature survey is rather modest. It is also highly recommended to mention for example this paper: B.S. Thomas et al., Strength and durability characteristics of copper tailing concrete. Construction and Building Materials, vol. 48 (2013), pp. 894-900. It has a very similar, if not identical, topic. This can be treated as a starting point to demonstrate the added value of your paper.
Author Response
Comments and Suggestions for Authors
1、It is rather a lab test report but a nice piece of work. The paper title should be completed as follows: on the compressive strength and Portland cement mortar.
Reply:Thank you very much for your Suggestions. Based on my research and your Suggestions, I will modify the topic as“ Effect of the content of micro-active copper tailing on the strength and pore structure of cementitious materials”
2、Table 2 – a misspelling, should be cement not ceme.
Reply: Cement mortar mix proportion and performance of specimens
3、Conclusion 1: “further addition of tailings does not affect the compressive strength of concrete”. This statement is doubly unjustified. First of all, the subject of the study is not concrete but mortar. Secondly, the influence of the copper tailings contested above 30% has been not tested.
Reply: I have deleted the sentence "the further addition of tailings does not affect the compressive strength of concrete" and condensed the conclusion。
4、Conclusion 2: “speed of decrease” better “rate of decrease”. Proof-reading by a native speaker is strongly recommended.
Reply:Native English speakers have been invited to proofread the article.
5、The literature survey is rather modest. It is also highly recommended to mention for example this paper: B.S. Thomas et al., Strength and durability characteristics of copper tailing concrete. Construction and Building Materials, vol. 48 (2013), pp. 894-900. It has a very similar, if not identical, topic. This can be treated as a starting point to demonstrate the added value of your paper.
Reply:Thank you very much for recommending to me the articles of B.S. Thomas et al.The content of our research is very different: I mainly reveal the micro mechanism of the influence of copper tailings content on the compressive strength of cement mortar containing copper tailings, and then determine the optimal proportion of cementing materials containing copper tailings from the aspect of gradation adjustment.B.S. Thomas et al. studied the optimal content of copper tailings in concrete from a macro perspective.

Reviewer 2 Report
The manuscript presents the results of measurements concerning the effects of the various levels of addition of copper tailings on the compressive strength, microstructure development and pore size distribution of concrete/mortar up to 365 days of hydration.
From the analysis of the presented manuscript I found the following:
1) Title: I would suggest to consider modifying the title as the Authors did not study only cement but materials based on cement (concrete/mortars/cementitious materials).
2) Abstract: The Authors should decide how they want to call the material they studied. If it is “cement slurry”, “cementitious material”, “copper-tailing composite cementitious material”, “composite cementitious material”, “cement-copper tailing composite cementitious material slurry”, “copper tailing composite cementitious materials.” The abstract is confusing if the material is not properly defined. I would suggest to name it as “cementitious material containing copper tailings” and in the following text, I would use only shortcut of CMCT to avoid repetition.
3) Abstract: “Results showed that with increasing copper tailing content, the compressive strength was lower, porosity was higher, and pore diameter was greater (before a curing age of 365 d under standard conditions). The compressive strength of the composite cementitious material increased, whereas its porosity and pore diameter decreased with increasing copper tailing content and longer curing time.” These two sentences are in disagreement. The first one states that the increasing amount of copper tailing leads to increase of porosity which leads to decrease of compressive strength, whereas the following sentence claims the opposite.
4) Abstract + whole text: The hardened cementitious material/concrete/mortar containing copper tailing should not be called slurry.
5) Abstract: “The micro-aggregate filling of copper tailings is primarily responsible for the strength increase of the copper tailing composite cementitious materials.” In later ages of hydration or when and based on what?
6) Keywords: The authors never used “copper tail slag”, and therefore, this keyword is inaccurate.
7) Introduction: “Domestic [5–8] and international experts have focused on the preparation of concrete as a mineral admixture and studied its performance.” Concrete is not a mineral admixture. Also a more specific review of the results from the cited papers + more up-to-date references should be provided.
8) Introduction: “Research showed that the pore structure has a significant impact on the mechanical properties and carbonation resistance [9], chloride ion erosion resistance [10], and other durability properties of concrete [11]. Did you mean “ordinary concrete”? if yes, I would specify it in the text.
9) Figure 1: the height of the pictures should be kept the same.
10) Table 1: The results of both, cement and copper tailing slag should be provided with the same significant digits.
11) Figure 2: Copper tailings is not cementitious material. There is no label a) and b) on the figures. Why the cumulative curves are not provided in the same format? It would be more clear if the cumulative curves of the studied materials are provided in one figure and the particle size distribution curves in another.
12) Raw materials: The standard sand with what particle size distribution? It should be included in Figure 2 as well.
13) Raw materials: Description of Figure 2 contains references [15,16], which are not included in the list of references. Why these specific results are being supported by the references?
14) Raw materials: “The particle size distribution of the cement is reasonable.” It should be explained why the PSD of cement is “reasonable”.
15) Production of test pieces: The methods providing the results of the initial and final setting time + fluidity should be mentioned.
16) Production of test pieces: In this part, the studied materials are called as mortars whereas here it would be appropriate to call them slurry actually.
17) Production of test pieces: this part of the manuscript should be written in the past tense.
18) Compressive strength test: How many samples were used for the compressive strength tests? The standard deviation of the results should be provided in Figure 3.
19) SEM observations: What was the magnification and were the sample specially treated before analysis?
20) Results and discussion: Compressive strength results are written in another type of font. The explanations of the results are not clear. For example: “The compressive strength of each specimen increases with increasing curing age. With copper tailing content increase , when the curing age is less than 180 d, the compressive strength of each mortar specimen decreases, whereas when the curing age exceeds 180 d, the compressive strength of mortar sample increases sharply.” Based on Figure 3, the compressive strength is generally lower with increasing amount of copper tailing in the structure of studied materials but it definitely does not decrease within the studied time period. The obtained results should be compared to the literature. The following discussion of the results is only a hypothesis, which should be supported by another results. In addition, another labelling of samples is provided, as compared to the Table 2. The same problem occurs in Figure 5.
21) Figure 4: Again, the labels a-c) are missing on the pictures. Moreover, every picture has different dimensions. The dark areas are pores? Isn’t it the opposite?
22) MIP results for hole structure: Hole structure or whole?
23) Porosity: The first paragraph does not follow the same formatting like the rest of the manuscript. The description of the results is not clear. Again the issue with “slurry”. What is the difference between porosity and the porosity rate?
24) Maximum aperture distribution: Maximum pore size distribution? What kind of “many early hydration products”? The discussed results should be supported with the literature. Based on what the Authors deduce that the capillary pores decrease due to filling by the copper tailing slag?
25) Figure 6: Again, the labelling is missing. The size of figures should be kept the same.
26) Figure 7: Missing labelling. The magnification of the figures is not the same. The size of the figures should be kept the same. Anything else than Ca(OH)2 and copper tailing was found by means of SEM analysis?
27) Conclusions: It is only a repetition of the previously presented results with no additional information.
28) Generally, the style (fonts) of the figures should be kept the same.
29) Acknowledgements: EDX and XRD results were not provided in this study.
On the basis of the specified ones, I consider that the manuscript can only be published after major changes will be made to the content of the manuscript.
Author Response
From the analysis of the presented manuscript I found the following:
1) Title: I would suggest to consider modifying the title as the Authors did not study only cement but materials based on cement (concrete/mortars/cementitious materials).
Respond:Thank you very much for your Suggestions. Based on my research and your Suggestions, I modify the title as follows: influence of micro-active copper tailings slag content on compressive strength and pore structure of cementing materials containing copper tailing.
2) Abstract: The Authors should decide how they want to call the material they studied. If it is “cement slurry”, “cementitious material”, “copper-tailing composite cementitious material”, “composite cementitious material”, “cement-copper tailing composite cementitious material slurry”, “copper tailing composite cementitious materials.” The abstract is confusing if the material is not properly defined. I would suggest to name it as “cementitious material containing copper tailings” and in the following text, I would use only shortcut of CMCT to avoid repetition.
Respond:According to your suggestion, the cementing materials containing copper tailings have been uniformly used to represent the materials studied.
3) Abstract: “Results showed that with increasing copper tailing content, the compressive strength was lower, porosity was higher, and pore diameter was greater (before a curing age of 365 d under standard conditions). The compressive strength of the composite cementitious material increased, whereas its porosity and pore diameter decreased with increasing copper tailing content and longer curing time.” These two sentences are in disagreement. The first one states that the increasing amount of copper tailing leads to increase of porosity which leads to decrease of compressive strength, whereas the following sentence claims the opposite.
Respond: at the early stage of curing time,with increasing copper tailing content, the compressive strength of cement mortar with copper tailing (CMCT) was lower, porosity and pore diameter of CPCT were higher and greater;with the extension of curing age, when the content of copper tailing was less than 30%, the compressive strength of CMCT and the porosity of CPCT changed slightly with the increase of the content of copper tailing.
4) Abstract + whole text: The hardened cementitious material/concrete/mortar containing copper tailing should not be called slurry.
Respond: According to the actual situation, it is modified into mortar containing copper tailings or hardening cementing materials containing copper tailings.
5) Abstract: “The micro-aggregate filling of copper tailings is primarily responsible for the strength increase of the copper tailing composite cementitious materials.” In later ages of hydration or when and based on what?
Respond: In the later stage of cement hydration,the micro-aggregate filling of copper tailing is primarily responsible for the strength increase of the CMCT.
6) Keywords: The authors never used “copper tail slag”, and therefore, this keyword is inaccurate.
Respond: Copper tailing slag modified to copper tailing.
7) Introduction: “Domestic [5–8] and international experts have focused on the preparation of concrete as a mineral admixture and studied its performance.” Concrete is not a mineral admixture. Also a more specific review of the results from the cited papers + more up-to-date references should be provided.
Respond: I am so sorry ,this is my Writing errors.Domestic and international experts[5–8] have focused on the preparation of concrete as a mineral admixture and studied its performance.
8) Introduction: “Research showed that the pore structure has a significant impact on the mechanical properties and carbonation resistance [9], chloride ion erosion resistance [10], and other durability properties of concrete [11]. Did you mean “ordinary concrete”? if yes, I would specify it in the text.
Respond: yes ,it is ordinary concrete.
9) Figure 1: the height of the pictures should be kept the same.
Respond: The height of theFigure 1 has been modified consistently
10) Table 1: The results of both, cement and copper tailing slag should be provided with the same significant digits.
Respond: The significant bits have been unified.
11) Figure 2: Copper tailings is not cementitious material. There is no label a) and b) on the figures. Why the cumulative curves are not provided in the same format? It would be more clear if the cumulative curves of the studied materials are provided in one figure and the particle size distribution curves in another.
Respond: As you suggested, the cumulative curve is represented by one graph, the particle size distribution curve by another graph.
12) Raw materials: The standard sand with what particle size distribution? It should be included in Figure 2 as well.
Respond: As you suggested, I add particle size distribution of standard sand .
13) Raw materials: Description of Figure 2 contains references [15,16], which are not included in the list of references. Why these specific results are being supported by the references?
Respond:Corresponding literature has been supplemented
14) Raw materials: “The particle size distribution of the cement is reasonable.” It should be explained why the PSD of cement is “reasonable”.
Respond:corresponding literature has been supplemented
15) Production of test pieces: The methods providing the results of the initial and final setting time + fluidity should be mentioned.
Respond:Corresponding literature has been supplemented.
16) Production of test pieces: In this part, the studied materials are called as mortars whereas here it would be appropriate to call them slurry actually.
Respond:The cement slurry is modified to cement paste containing copper tailing
17) Production of test pieces: this part of the manuscript should be written in the past tense.
Respond: The tense has been modified.
18) Compressive strength test: How many samples were used for the compressive strength tests? The standard deviation of the results should be provided in Figure 3.
Respond:Six samples were used.
19) SEM observations: What was the magnification and were the sample specially treated before analysis?
Respond:The target age hardened paste was taken, the volume was broken to about 1 cm3, hydration was terminated by anhydrous ethanol, and vacuum dried at 60℃ until constant weight was used for backup.the amplification factor was500times.
20) Results and discussion: Compressive strength results are written in another type of font. The explanations of the results are not clear. For example: “The compressive strength of each specimen increases with increasing curing age. With copper tailing content increase , when the curing age is less than 180 d, the compressive strength of each mortar specimen decreases, whereas when the curing age exceeds 180 d, the compressive strength of mortar sample increases sharply.” Based on Figure 3, the compressive strength is generally lower with increasing amount of copper tailing in the structure of studied materials but it definitely does not decrease within the studied time period. The obtained results should be compared to the literature. The following discussion of the results is only a hypothesis, which should be supported by another results. In addition, another labelling of samples is provided, as compared to the Table 2. The same problem occurs in Figure 5.
Respond:The number of samples has been unified, the font has been unified, and the wrong sentence has been corrected.
21) Figure 4: Again, the labels a-c) are missing on the pictures. Moreover, every picture has different dimensions. The dark areas are pores? Isn’t it the opposite?
Respond:Figure 4:Labels a-c) have been added, the dimensions have been unified, and the white tends to be pores.
22) MIP results for hole structure: Hole structure or whole?
Respond:Hole structure
23) Porosity: The first paragraph does not follow the same formatting like the rest of the manuscript. The description of the results is not clear. Again the issue with “slurry”. What is the difference between porosity and the porosity rate?
Respond:The cement mortar and cement slurry containing copper tailing have been numbered and the format of the paper has been revised.Porosity is mainly for hardened copper - bearing tailing cement slurry.
24) Maximum aperture distribution: Maximum pore size distribution? What kind of “many early hydration products”? The discussed results should be supported with the literature. Based on what the Authors deduce that the capillary pores decrease due to filling by the copper tailing slag?
Respond:Two additional articles have been added to support the results.
25) Figure 6: Again, the labelling is missing. The size of figures should be kept the same.
Respond:Added tags and changed the size of the Numbers
26) Figure 7: Missing labelling. The magnification of the figures is not the same. The size of the figures should be kept the same. Anything else than Ca(OH)2 and copper tailing was found by means of SEM analysis?
Respond:The size of the Numbers is uniform. No.
27) Conclusions: It is only a repetition of the previously presented results with no additional information.
Respond:The conclusion of this paper is condensed
28) Generally, the style (fonts) of the figures should be kept the same.
Respond:The corresponding graph has been modified
29) Acknowledgements: EDX and XRD results were not provided in this study.
Respond: The EDX and XRD results are deleted
On the basis of the specified ones, I consider that the manuscript can only be published after major changes will be made to the content of the manuscript.

Round 2
Reviewer 2 Report
The Authors have addressed in detail every comment I made and they have provided the relevant data needed. I have no further comments.